# Insights into cultural and compliance challenges in type 2 diabetes care: A qualitative study of Moroccan and Belgian patients in Belgium

**Stefaan Six**[1]*, **David Israel**[1], **Johan Bilsen**[1‡], **Aan Kharagjitsing**[2‡]

**1** Mental Health & Wellbeing research group, Department of Public Health, Vrije Universiteit Brussel, Brussels, Belgium, **2** Department of Endocrinology & Diabetology, Universitair Ziekenhuis Brussel, Brussels, Belgium

‡ JB and AK share last authorship on this work.
* Stefaan.Six@vub.be

**Data Availability Statement:** All relevant data are within the manuscript and its Supporting information files.

## Abstract

### Aims

To explore factors that may contribute to a possible reduced compliance in patients with type 2 diabetes mellitus (T2DM) with a migrant (i.e. North African) background living in a western society.

### Methods

Semi-structured interviews with people with T2DM both of Moroccan and Belgian origin, recruited within the diabetes clinic of the University Hospital Brussel, Belgium. Data was analysed thematically using NVivo.

### Results

Participants indicated they were adequately informed about T2DM, however results show a demand for culturally tailored preventive education for Moroccan participants. Both groups generally had good knowledge of a healthy lifestyle and what is expected after diagnosis, but considered maintaining healthy lifestyle and correct medication adherence, intensive. Participants mentioned a wide range of themes that affected their compliance, both positively and negatively. Perceived barriers were social issues, lack of motivation, insufficient support from the environment, stress, forgetfulness, winter conditions and COVID. Culturally shaped views on eating habits, illness, medication use and health were clear barriers in the Moroccan group.

### Conclusion

Findings highlight the need for future in depth research into diabetes related knowledge within the Moroccan community living in Belgium (and similar other Western countries)

**Funding:** The author(s) received no specific funding for this work.

**Competing interests:** The authors have declared that no competing interests exist.

whilst considering differences between generations of migrants, gender and level of education.

## Introduction

In 2021, an estimated 536.6 million people worldwide suffered from diabetes, a chronic disease considered to be one of the top ten causes of death worldwide [1]. Moreover, within Europe, type 2 diabetes mellitus is considered one of the main causes of morbidity and mortality [2]. A higher prevalence of type 2 diabetes mellitus is noted among populations with a migrant background residing in developed western countries [2–5]. Immigration, as a social determinant of health, has been underexplored in public health research despite its significant impact on health outcomes [6].

Also in Belgium, a higher prevalence of type 2 diabetes mellitus was observed among patients of Moroccan and Turkish origin [7]. The existence of a clear link between this higher prevalence of type 2 diabetes mellitus and lower compliance in these population groups was also observed [5, 7]. The socio-cultural background of individuals within these populations may have a unique influence on how diabetes is viewed and managed. Research into the perspective of these patients on guidance, treatment and therapy compliance in diabetes can therefore provide useful indicators to promote therapy compliance [5]. However, research into patients' views on compliance with therapy recommendations in these ethnic populations is limited [8].

The aim of this study is to explore underlying factors that may lead to lower compliance in persons with diabetes of Moroccan origin living in a Western society. We focus on how people, both of Moroccan and Belgian descent, experience the management and treatment of diabetes. This can provide important insights to promote therapy compliance [5]. Better compliance is associated with improved clinical outcomes, improved health, reduced healthcare utilization and, ultimately, reduced healthcare costs [9].

## Material and methods

### Design

We used an exploratory qualitative design: open-ended, semi-structured interviews were conducted and recorded. Responses to each question were qualitatively analysed to determine themes. This study was approved by the medical ethics committee of the Vrije Universiteit Brussel and University Hospital (VUB &UZ-Brussel) and took place within the diabetes clinic of University Hospital (UZ) Brussel during the period from 6 April 2021 to 21 April 2021. Written informed consent to participate in the study and publish the results was obtained from all participants, whilst the study protocol was positively reviewed by the medical ethical institutional review board (approval number B.U.N. 1432021000375)).

### Recruitment of participants

Participants were recruited via purposive sampling. Inclusion criteria were: 1. having type 2 diabetes (as defined by ADA criteria) [10], 2. aged $\geq$ 18 years, 3. registered at the diabetes clinic for at least 6 months, and 4. being of Belgian or Moroccan descent. The recruitment of participants took place within the diabetes clinic of UZ Brussel, Belgium. Medical specialists responsible for consultations were informed about the study and referred participants based

on the inclusion criteria to the researcher who performed the interview (DI). During sample selection we aimed to include a similar number of people of Belgian and Moroccan origin, as well as aiming for a gender balance.

## Semi-structured interviews

An interview guide was compiled, inspired by the themes found by Peeters et al. (2015) who conducted a similar study with a different immigrant population [5]. The final interview guide was reviewed by SS and AK. Prior to data collection, the interview guide was tested by DI, SS and AK. In an experiential learning setting, DI was mentored by SS, a medical anthropologist, in the acquisition and application of cross-cultural communication skills. This was achieved through a role-play exercise, wherein AK, a diabetes specialist, took on the role of a typical patient. This scenario emulates the consultations that AK frequently conducts within his clinical practice. The relevance of this prior training is underlined by the fact that Brussels, the location of AK's practice, is the second most culturally diverse city globally. Statbel (2022), the Belgian statistical office, estimates that approximately 75% of the population in Belgium's capital comprises individuals of non-Belgian origin, with the top three countries of origin being Morocco, France, and Italy [11]. DI conducted all seventeen interviews; six were conducted in Dutch, with the others conducted in French (DI is fluent in both languages). The interviews took place in a designated room within the diabetes clinic and were audio recorded.

## Analysis

All interviews were recorded and transcribed verbatim. French-spoken interviews were transcribed and then translated into Dutch. We used thematic analysis as described by Clarke & Braun [12]. All transcripts were read closely for surface and underlying meaning and codes were developed to represent units of meaning. Primary coding was conducted by DI, reviewed by SS and discrepancies were resolved to consensus. Themes were developed by identifying patterns of meaning within and across transcripts and compared with the themes found by Peeters et al. [5]. Although the interview guide was inspired by the themes found by Peeters et al., we used primarily an inductive approach to coding and analysis, from the point of view of checking whether the same themes were found, as well as to allow new findings specific to the patient group of interest. Memo-writing was used to (further) develop themes and verify relationships between themes and data elements. These memos were discussed at team meetings to facilitate an integrated reflection on our relationship to the data, study participants and analysis. The final structure of relationship between themes and subthemes was decided by consensus of all investigators involved. Representative quotes were selected to illustrate themes. NVivo software package was used for coding and managing the data [13].

## Results

A total of 17 interviews were conducted; 8 with people of Belgian descent and 9 with people of Moroccan descent (see Table 1 for the demographic characteristics of the participants). The median interview duration was 32 minutes and 34 seconds. To improve the readability of the manuscript, in what follows we refer to people of Moroccan descent living in Belgium as 'Moroccans' and people of Belgian descent as 'Belgians'.

The analysis yielded 3 major themes: beliefs about diabetes, patient-doctor relationship and barriers/facilitators. See S1 Table for a complete list of all (sub)themes and supporting quotes.

Table 1. Participant demographic characteristics.

|  | Participants of Belgian descent | Participants of Moroccan descent |
|---|---|---|
| **Total number** | 8 | 9 |
| **Age** | | |
| 30–39 years | 0 | 1 |
| 40–49 years | 1 | 1 |
| 50–59 years | 1 | 2 |
| 60–69 years | 3 | 3 |
| 70–79 years | 2 | 1 |
| 80–89 years | 1 | 1 |
| **Sex** | | |
| Men | 4 | 4 |
| Women | 4 | 5 |
| **Immigrant generation** | | |
| First-generation immigrant | / | 7 |
| Second-generation immigrant | / | 1 |
| Third-generation immigrant | / | 1 |
| **Educational level** | | |
| None | 0 | 2 |
| Only primary education | 1 | 2 |
| Secondary education | 4 | 3 |
| Tertiary education | 3 | 2 |
| **Time since diagnosis** | | |
| 1–5 years | 2 | 3 |
| 6–10 years | 1 | 2 |
| 11–20 years | 4 | 3 |
| > 20 years | 3 | 2 |
| **Hypoglycaemic medication** | | |
| Only oral | 4 | 4 |
| Only insulin | 4 | 1 |
| Both | 0 | 4 |

## Beliefs about diabetes

All participants, except for one Moroccan patient and one Belgian patient, indicated that they received sufficient information about their illness and diagnosis. Two participants were convinced that diabetes is an acute illness. Three Belgian and four Moroccan participants stated that although they were well informed, they would have preferred to receive more preventive information. Moroccan participants highlighted that preventive information should be adapted to the Moroccan target group. Most participants, from both target groups, showed awareness that an unhealthy lifestyle can lead to diabetes.

> Yes, it is a question of education! What are they going to give Moroccans as good advice? They hardly know anything about it! But it is not their fault! There are many who cannot read or write.
>
> (Male, Moroccan)

Among Moroccan participants, a healthy lifestyle is more linked to avoiding sweets rather than having sufficient physical activity. Whereas among the Belgian group, healthy food in combination with physical activity is always seen as the ideal lifestyle. Five Moroccan participants did not deem it necessary to modify their eating habits on account of their disease.

I think that doesn't bother me—I still continue to appreciate [*unadjusted*] Moroccan cuisine which is very rich.

(Female, Moroccan)

Some Moroccan patients said they compensated for their unhealthy eating habits by adding extra vegetables to dishes. Two Moroccan participants mentioned that younger generations are better able to adapt their eating habits, while the older ones tend to be more conservative regarding nutritional changes.

There does seem to be a difference in lifestyle between generations. In general, the older generations are more used to traditional Moroccan cuisine. And the younger ones are more flexible and eat a more European cuisine

(Female, Moroccan).

Just over half of Moroccan participants assumed that diabetes has a hereditary factor. For five Moroccan participants, the onset of diabetes was linked to a stressful or emotional event. Among Belgian participants, religion had little influence on the perception of the disease, while three Moroccan participants were convinced that it was their fate, according to their religious beliefs, that they were diagnosed with diabetes. Some Moroccan participants indicated that they had no fear of the disease (T2DM) since several people close to them have grown old despite this condition. It is therefore no longer seen as something exceptional within the Moroccan community according to the participants.

It is also just assumed that the disease diabetes is something normal within our community!

(Female, Moroccan)

Proper compliance with prescribed therapy was experienced to be intensive by most participants. Two Moroccan participants expressed reluctance towards blood glucose self-monitoring (via finger prick testing). They would like to replace this by flash or continuous glucose monitoring (via a sensor worn on the body) but cited the financial aspect as a barrier to this switch. (. . .) If there is a way to get a sensor or something that measures blood without needles. They said it exists, but you must pay for it and it's expensive. I can't afford it. And I regret that. (Male, Moroccan) (*As of July 1$^{st}$ 2023 continuous glucose sensor (GCS) monitoring use is being reimbursed in Belgium for type 2 diabetes on an intensive insulin regimen (i.e. 3 or more injections per day); at the time of this study there was no reimbursement for CGS.*)

### Patient—Doctor relationship

Participants indicated high confidence in the expertise of both the general practitioner and the diabetologist, noting that they did not expect their general practitioner to be an expert on diabetes. In this sense, patients indicated more confidence in the diabetologist's

recommendations. This trust led patients to passively accept treatment recommendations, i.e., the majority of patients did not intend to check the validity of the recommendations made in any way. Only a proportion of participants indicated that they actively sought additional information about the proposed treatment plan. Patients indicated that the overall positive relationship with the general practitioner and diabetologist was partly due to the facilitation of communication in Dutch, French or, if necessary, in the patient's original language with the support of an interpreter.

> He is French speaking but can speak Dutch well and for medical matters you like to have information in your own language. It depends from hospital to hospital. You do like to have your own language when it comes to medical matters. Language does a lot because you understand the difficult terms better.
>
> (Female, Belgian)

With regard to the transfer of information between the relevant physician and the patient, none of the patients indicated that they experienced problems in the transfer of information. Moroccan patients indicated that although they have confidence in the professional's expertise, and therefore feel obliged to adhere to treatment as much as possible, this in itself is not sufficient to increase compliance. This mainly concerned nutrition and the proper use of medication.

> He has an influence: yes and no. He asks me to do a lot of things and then sometimes I think when I want to go and eat a burger, I am not going to do it anyway to satisfy the doctor. But sometimes I also think the opposite and really don't care what the doctor is going to say. And then I eat less healthy
>
> (Female, Moroccan).

Furthermore, two Belgian participants indicated that the information offered in and by the hospital could be more comprehensive. In addition, a Moroccan participant reported a lack of in-depth and detailed information, as only the general causes of diabetes were discussed with the diabetes specialist, without addressing the specific causes and details related to the individual diagnosis. Finally, one Moroccan participant reported withholding information about eating habits to maintain a false image of correctly following the treatment plan for fear of disappointing the physician.

## Barriers and facilitators

From the group of Belgian participants, only two persons indicated that they completely and correctly followed the proposed recommendations. This was not the case in the Moroccan group where none of the participants stated to be fully compliant.

For both groups, maintaining a balanced and healthy intake of food seemed to be challenging, albeit for different reasons. The Belgian participants, indicated to experience difficulties in social settings whereas the Moroccan group, indicated to experience difficulties given that the required changes are difficult to combine with the traditional Moroccan kitchen. The latter may be related to the dominant male culture, in which Moroccan women often conform to their husbands' wishes, which do not always correspond to the advised nutritional plan.

Women are usually submissive; the man is usually the boss in the house. The woman usually must listen to the man's wishes, their life is more difficult. (. . .) And this can then have an effect on bad living habits.

(Female, Moroccan)

From a religious point of view the vast majority indicated that their religion and more specifically strict fasting during Ramadan, does not pose a barrier for adhering to treatment as their religion in principle exonerates ill people of the requirement to fast.

Since I am diabetic, I am considered someone who is sick. So, I have the right to eat. So, I do not participate in Ramadan. So, I don't adjust my medication either.

(Female, Moroccan)

Belgian participants who, as discussed, experience difficulties in social settings also indicate to experience difficulties because of both practical considerations and lack of support from the environment.

You have to leave a lot behind. You also have to have the character and support of the people around who help you to do that. Having support helps a lot. That you are not allowed to snack anymore. (. . .) My husband likes to eat biscuits for breakfast. I dare to eat two pastries then.

(Female, Belgian)

Note that the latter has been indicated to be both a barrier as well as a main facilitator depending on the participant. In addition, mainly the Belgian participants indicated to be able to adhere better to treatment as a direct result of their retirement.

In both groups, a lack of intrinsic motivation appears to be at the basis of limited physical activity. In the group of Belgian, things such as a lack of time, being physically unable to exercise and having done enough sport in the past are mentioned as reasons for not exercising enough. Moroccan participants considered adequate physical activity to be less important in their treatment plan.

Taking treatment less seriously also appears to lead to less consistent compliance with medication intake. Participants who took the disease too lightly showed little interest in taking their medication correctly.

In the beginning, I was a bit sceptical, I didn't believe in it very much. I thought it was a simple disease like a cold with no consequences. (. . .) That's why I didn't have much interest in taking my medicines. When I forgot to take them, I didn't think it was serious.

(Female, Belgian)

Some Moroccan participants described adjusting medication on their own based on their meals. When participants ate a healthy meal, doses of medication were skipped. Forgetfulness due to fatigue, the amount of medication or breaking a certain routine were also cited by participants for not taking their medication correctly.

At 22h15, the long-term insulin was then injected. And I did have problems with that. Sometimes I was asleep before I got there

(Male, Belgian).

Among Moroccan participants, scepticism towards medication, expected outcomes and actual results showed reduced compliance.

One day I do it, two days I don't and that's it. I tell myself it's always the same. I see the results. So, there's no point.

(Male, Moroccan).

This in combination with the fact that they indicated to freely adjust the medication and treatment plan in function of their lifestyle at a current point in time as they saw fit. A last frequently mentioned barrier within the Moroccan group was 'stress'. Five participants reported that they found it very challenging to follow the proposed therapy during times of stress or emotional difficulties. This applied to both medication and lifestyle.

As soon as I have a moment of mental difficulty and go into a slump, I spend less attention on the disease.

(Female, Moroccan)

In both groups, patients reported that adhering to therapy is supported by having structure and routine.

You get used to that (. . .). That becomes a routine in the morning, I get up 15 minutes early for that. I put them in a bag from the supermarket here. Yes, every morning I just take those. The ones from the evening are on my night table and before going to bed I take them. I actually have little problem taking that. I rarely skip that.

(Male, Belgian)

Additionally, all participants expressed a desire for improved health and acknowledged that they adhered to treatment out of fear of the consequences of non-compliance. However, some patients did not consider the disease to be serious, and as a result, they gave less importance to treatment compliance.

## Discussion

In the present study, we employed a qualitative methodology to explore the potential impact of treatment-related beliefs and other factors on therapy compliance among type 2 diabetes patients, with a specific emphasis on comparing patients of Moroccan origin to patients of Belgian origin. The aim was to elucidate any disparities between these two groups.

Our findings align with previous research; specifically, other studies have identified similar key factors influencing compliance in diabetes patients from migrant backgrounds, including socio-economic status, education, beliefs about illness, medication, and lifestyle; forgetfulness; cultural traditions; social support; the doctor-patient relationship; climate; religion; and mental health [5, 8, 14–19].

In our study, we assessed patients' inherent beliefs regarding overall treatment in conjunction with the patient-physician relationship, how information is provided and shared, and the barriers and facilitators experienced by patients from both groups and related this to perceived therapy compliance. In addition, we evaluated and discussed the differences observed between the Moroccan and Belgian participant groups.

The key finding in relation to the perceived therapy compliance is that for the participants of Moroccan descent, lack of adherence to treatment appears mainly justified through religion i.e., Moroccan patients tend to believe that their illness is ordained and therefore they are required to accept it. The resulting acceptance indirectly leads to a sub-par compliance with the recommended treatment plan. These findings are in line with the study of Snijder ea. (2017) who found that the main barriers to diabetes care among Moroccan migrants living in the Netherlands were related to ethnicity-specific characteristics (such as culture, migration history, ethnic identity, socioeconomic factors and discrimination) [20]. An additional finding in our study is that the normalization of the disease in Moroccan culture may be attributed, at least in part, to inadequate preventive information that may lead to a misperception of the disease's gravity. The misperception of the disease's severity contributes to a passive attitude among patients, leading them to question or neglect doctors' recommendations despite their trust in the doctor's expertise and the established interpersonal relationship. The third and final important finding pertains to non-adherence to the treatment plan due to the male-dominant culture in conjunction with traditional kitchen practices. Specifically, women tend to suppress their own need(s) for treatment adherence in order to comply with their spouse's wishes, whilst the traditional kitchen itself does not readily accommodate the necessary dietary changes required for proper treatment compliance. In addition, patients from Moroccan descent have shown to either increase or reduce medicine intake in coordination with their meal intake.

The above findings also hold to a certain extent for the Belgian participants with a lower emphasis on religion having an impact on treatment compliance. As such, aspects such as cultural habits, perception of the disease and required medication as well as the relation between compliance to treatment and, the patient-doctor relation are observed here as well. It can be inferred that patients with a Belgian cultural background may understandably be more receptive to Western medicine compared to patients with a non-Western cultural background. They tend to question their use of medication less and take more personal responsibility for both the cause of their condition and its improvement towards the future. This is in contrast with patients of Moroccan descent who tend to relate causality to their religious beliefs.

A distinct contrast is evident between the two groups in terms of how religion affects their illness experience. This shows how perceived treatment compliance is different in both groups and is (at least partly) driven by different beliefs and perceptions which are of key influence. As demonstrated in the theme on barriers and facilitators, patients within each of the subgroups perceived certain factors as either a barrier or a facilitator in their compliance to treatment based on their pre-existing personal beliefs, such as their religious framework. While this may lead to easier acceptance by allowing patients to shift causality and responsibility towards their religious convictions, it can also serve as a driver for compliance in some cases, such as during Ramadan where religion facilitates ill persons. These findings underscore the significant influence of personal prior beliefs and perceptions on the perceived compliance to treatment. Consequently, culturally appropriate diabetes education is necessary and, as earlier research has consistently shown, provides benefits over conventional care in terms of glycaemic control and diabetes knowledge [21].

## Strengths and limitations

This study has notable strengths, as it represents the first investigation of its kind in a Belgian context. Specifically, this research examines underlying factors, beliefs, and experiences that shape therapy compliance among an important ethnic minority group in Belgium which has

not been studied previously. The application of semi-structured interviews as a qualitative research method enabled patients to provide a comprehensive account of their experiences.

This study also has limitations. Firstly, the potential for generalization of these findings is limited due to the small sample size. It is unclear whether the findings can be applied to other contexts. However, the results could potentially be transferred to other Moroccan communities with similar characteristics, such as level of education, generation of immigrants, and location. Secondly, recruitment of participants through the doctors present in the diabetes clinic may potentially have introduced selection bias. The doctors made the decision to refer participants to the researcher for inclusion in the study, which could have influenced characteristics of the study population. Thirdly, in this study we did not collect socioeconomic data or variables related to possible systemic inequities, thereby limiting our ability to assess the potential role of non-culture related factors, such as e.g. healthcare access, or morbidity differences between the migrant and host population [22].

Despite these limitations, our findings show that cultural factors play a significant role in treatment compliance in this patient group and that it is necessary to include these elements to improve patient care. More research is needed in other health care settings with participants from the Moroccan community to further explore how care is better tailored to this group.

## Conclusion

The present study identified unique barriers and facilitators to treatment compliance among individuals of Moroccan descent residing in Belgium. The findings underscore the pressing need for targeted, culturally appropriate preventive information for this population. Specifically, information dissemination strategies should be tailored to reach individuals with varying levels of education, including the elderly, to ensure that accurate and comprehensive knowledge regarding diabetes is disseminated effectively. A key challenge for future research and future interventions is to develop strategies that are consistent with and respect cultural traditions, take into account additional potential influencing factors such as socioeconomic background while seeking to promote positive health outcomes. Thus, additional research is needed to assess the diabetes-related knowledge of the Moroccan diabetes patients in detail, identify barriers to preventive information transfer, and develop culturally sensitive educational interventions that effectively address these challenges within the Moroccan community.

## Supporting information

**S1 Table. Themes, subthemes and illustrative quotes.**
(DOCX)

## Author Contributions

**Conceptualization:** Stefaan Six, David Israel, Johan Bilsen, Aan Kharagjitsing.

**Data curation:** David Israel.

**Formal analysis:** David Israel, Aan Kharagjitsing.

**Investigation:** David Israel.

**Methodology:** Stefaan Six, David Israel, Aan Kharagjitsing.

**Project administration:** David Israel.

**Supervision:** Stefaan Six, Johan Bilsen, Aan Kharagjitsing.

**Writing – original draft:** Stefaan Six, David Israel, Aan Kharagjitsing.

**Writing – review & editing:** Stefaan Six, David Israel, Johan Bilsen, Aan Kharagjitsing.

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
