## [Decision Letter · Decision Letter 0]

13 Jun 2024

PONE-D-24-08750Insights into Cultural and Compliance Challenges in Type 2 Diabetes Care: A Qualitative Study of Moroccan and Belgian Patients in Belgium.PLOS ONE

Dear Dr. Six,

Thank you for submitting your manuscript to PLOS ONE. After careful consideration, we feel that it has merit but does not fully meet PLOS ONE’s publication criteria as it currently stands. Therefore, we invite you to submit a revised version of the manuscript that addresses the points raised during the review process.

We look forward to receiving your revised manuscript.

Kind regards,

Aziz Belkadi

Academic Editor

PLOS ONE

Journal Requirements:

Additional Editor Comments:

Kindly respond to the the reviewer's minor concerns. Thank you

Reviewers' comments:

Reviewer's Responses to Questions

**Comments to the Author**

1. Is the manuscript technically sound, and do the data support the conclusions?

Reviewer #1: Yes

2. Has the statistical analysis been performed appropriately and rigorously? 

Reviewer #1: N/A

3. Have the authors made all data underlying the findings in their manuscript fully available?

Reviewer #1: Yes

4. Is the manuscript presented in an intelligible fashion and written in standard English?

Reviewer #1: Yes

5. Review Comments to the Author

**Reviewer #1:** Six et al. provided insight into cultural differences that lead to disparity in adherence to Type 2 diabetes medications. The study used a very small sample size. Thus, the findings of this study cannot be generalized to a population group. The study's authors have highlighted these limitations at the end of the discussion section with other relevant shortcomings of the study, like no data about the participants' socioeconomic status was collected.

I have a couple of remarks:

1. The text can replace the reference to “God” with religious belief.

2. Did authors find any difference in adherence between first-generation Moroccans and second- and third-generation Moroccans?

6. PLOS authors have the option to publish the peer review history of their article (what does this mean?). If published, this will include your full peer review and any attached files.

Reviewer #1: No

---

## [Author Response · Author response to Decision Letter 0]

28 Jun 2024

Dear Dr. Belkadi,

Thank you for the opportunity to revise our manuscript. We have addressed the reviewer's points and your comments regarding the journal's requirements. (We have also uploaded a response to reviewers file that may be easier to read)

We hope our revised manuscript meets the standards.

With kind regards and appreciation for your voluntary work as editor,

Dr. Stefaan Six (on behalf of all authors)

Editor

Author reply: We have adjusted the manuscript so it complies with all PLOS ONE’s style requirements.

Author reply: We confirm that the Data Availability Statement is correct.

Author reply: NA (no images in the manuscript)

Author reply: We have reviewed the reference list for completeness and correctness.

Reviewer 1

1. The text can replace the reference to “God” with religious belief.

Author reply:

Thank you for your comment. We have replaced all references to “God” with a reference to religious belief.

2. Did authors find any difference in adherence between first-generation Moroccans and second- and third-generation Moroccans?

Author reply:

In our study, the primary intergenerational difference observed was in dietary habits, with older generations appearing to adhere more strictly to traditional Moroccan cuisine. This finding is detailed in the Results section under the theme “Beliefs about diabetes”. However, the sample included only one second-generation and one third-generation Moroccan immigrant. Due to this limited representation, we did not analyze differences in medication adherence between the generations within the Moroccan group. The small sample size would not have provided meaningful conclusions about generational differences in adherence. Instead, our focus was on the broader cultural and compliance challenges between Belgian and Moroccan groups. Future quantitative research with a larger sample size is necessary to address the valuable remark of the reviewer, i.e. to explore intergenerational differences in medication adherence among Moroccan immigrants more comprehensively.

---

## [Decision Letter · Decision Letter 1]

8 Sep 2024

Insights into Cultural and Compliance Challenges in Type 2 Diabetes Care: A Qualitative Study of Moroccan and Belgian Patients in Belgium.

PONE-D-24-08750R1

Dear Dr. Six,

We’re pleased to inform you that your manuscript has been judged scientifically suitable for publication and will be formally accepted for publication once it meets all outstanding technical requirements.

Kind regards,

Aziz Belkadi

Academic Editor

PLOS ONE

Additional Editor Comments (optional):

The authors have responded to the reviewer's concerns. The manuscript looks fine for me. Thank you!

Reviewers' comments:

Reviewer's Responses to Questions

**Comments to the Author**

1. If the authors have adequately addressed your comments raised in a previous round of review and you feel that this manuscript is now acceptable for publication, you may indicate that here to bypass the “Comments to the Author” section, enter your conflict of interest statement in the “Confidential to Editor” section, and submit your "Accept" recommendation.

Reviewer #1: All comments have been addressed

2. Is the manuscript technically sound, and do the data support the conclusions?

Reviewer #1: Yes

3. Has the statistical analysis been performed appropriately and rigorously? 

Reviewer #1: N/A

4. Have the authors made all data underlying the findings in their manuscript fully available?

Reviewer #1: Yes

5. Is the manuscript presented in an intelligible fashion and written in standard English?

Reviewer #1: Yes

6. Review Comments to the Author

Reviewer #1: The authors have addressed all the comments satisfactory and the manuscript is updated accordingly.

7. PLOS authors have the option to publish the peer review history of their article (what does this mean?). If published, this will include your full peer review and any attached files.

Reviewer #1: No

---

## [Editor Report · Acceptance letter]

15 Sep 2024

PONE-D-24-08750R1 

PLOS ONE

Dear Dr. Six, 

I'm pleased to inform you that your manuscript has been deemed suitable for publication in PLOS ONE. Congratulations! Your manuscript is now being handed over to our production team.

Kind regards, 

on behalf of

Dr. Aziz Belkadi 

Academic Editor

PLOS ONE